# A Tryptophan ‘Gate’ in the CRISPR-Cas3 Nuclease Controls ssDNA Entry into the Nuclease Site, That When Removed Results in Nuclease Hyperactivity

**DOI:** 10.3390/ijms22062848

**Published:** 2021-03-11

**Authors:** Liu He, Zoe Jelić Matošević, Damjan Mitić, Dora Markulin, Tom Killelea, Marija Matković, Branimir Bertoša, Ivana Ivančić-Baće, Edward L. Bolt

**Affiliations:** 1School of Life Sciences, University of Nottingham, Nottingham NG7 2UH, UK; Liu.He1@nottingham.ac.uk (L.H.); Tom.Killelea@nottingham.ac.uk (T.K.); 2Department of Chemistry, Faculty of Science, University of Zagreb, 10000 Zagreb, Croatia; zoe.jelic.matosevic@chem.pmf.hr (Z.J.M.); bbertosa@chem.pmf.hr (B.B.); 3Department of Biology, Faculty of Science, University of Zagreb, 10000 Zagreb, Croatia; damjan.mitic@biol.pmf.hr (D.M.); dora.markulin@biol.pmf.hr (D.M.); 4Institute Ruđer Bošković, 10000 Zagreb, Croatia; Marija.Matkovic@irb.hr

**Keywords:** helicase, Cas3, CRISPR, genome editing

## Abstract

Cas3 is a ssDNA-targeting nuclease-helicase essential for class 1 prokaryotic CRISPR immunity systems, which has been utilized for genome editing in human cells. Cas3-DNA crystal structures show that ssDNA follows a pathway from helicase domains into a HD-nuclease active site, requiring protein conformational flexibility during DNA translocation. In genetic studies, we had noted that the efficacy of Cas3 in CRISPR immunity was drastically reduced when temperature was increased from 30 °C to 37 °C, caused by an unknown mechanism. Here, using *E. coli* Cas3 proteins, we show that reduced nuclease activity at higher temperature corresponds with measurable changes in protein structure. This effect of temperature on Cas3 was alleviated by changing a single highly conserved tryptophan residue (Trp-406) into an alanine. This Cas3^W406A^ protein is a hyperactive nuclease that functions independently from temperature and from the interference effector module Cascade. Trp-406 is situated at the interface of Cas3 HD and RecA1 domains that is important for maneuvering DNA into the nuclease active site. Molecular dynamics simulations based on the experimental data showed temperature-induced changes in positioning of Trp-406 that either blocked or cleared the ssDNA pathway. We propose that Trp-406 forms a ‘gate’ for controlling Cas3 nuclease activity via access of ssDNA to the nuclease active site. The effect of temperature in these experiments may indicate allosteric control of Cas3 nuclease activity caused by changes in protein conformations. The hyperactive Cas3^W406A^ protein may offer improved Cas3-based genetic editing in human cells.

## 1. Introduction

Prokaryotes have evolved multiple defence strategies against incursion from mobile genetic elements (MGEs, e.g., bacteriophage), including activation of their CRISPR-Cas systems. These comprise a DNA locus called ‘CRISPR’ (Clustered Regularly Interspaced Short Palindromic Repeats) and genes encoding ‘Cas’ (CRISPR associated) proteins, and sometimes additional non-Cas proteins. Comprehensive recent perspectives of CRISPR systems can be found in [1,2,3], and the references therein. Functional interaction and co-operation between CRISPR DNA and its transcribed RNA (crRNA) and Cas proteins generate immunity against future infection by the same or similar MGEs (CRISPR ‘Adaptation’) and destroy the MGE (CRISPR ‘Interference’). For the latter, two major activities are needed—an effector ribonucleoprotein complex to target crRNA to MGE DNA or RNA via base-pairing, and a nuclease that degrades the MGE. Interference reactions in class 1 CRISPR-Cas systems such as that of *E. coli* — CRISPR classifications can be found in [4]—are catalyzed by Cascade complex and the Cas3 nuclease-helicase that are both essential for efficacious defence against MGEs [5]. Cascade scans duplex DNA until it encounters a trinucleotide sequence termed a protospacer adjacent motif (PAM) that stabilizes it into a ‘seed’ complex [6,7,8,9,10,11]—PAMs and other nucleic acid motifs involved in the functioning of CRISPR systems are reviewed in [12]. The composition of the PAM sequence that is recognized by any given Cascade complex is determined from the CRISPR adaptation reaction that builds immunity, by inserting an MGE ‘protospacer’ DNA into a CRISPR locus in a specific orientation—adaptation processes were reviewed recently in [13,14,15]. Recognition of PAM and the ‘seed’ reaction by Cascade trigger formation of an R-loop (‘RNA-loop’) complex in which Cascade base-pairs crRNA with a ‘targeted’ strand of DNA duplex, and the other ‘non-targeted’ DNA strand is displaced as ssDNA [16]—the importance of R-loops in biology is reviewed recently in [17]. In CRISPR systems, Cas3 is recruited to the Cascade R-loop complex and degrades DNA, processes reviewed recently in [18]. Therefore, Cas3 destroys MGE DNA and in the process generates DNA fragments that may prime adaptation [19], by their capture and integration into CRISPR DNA, completing a virtuous cycle of CRISPR activities.

Control of the Cas3 nuclease-helicase activity is exerted at several junctures. Transcription of *E. coli ygcB* (encoding Cas3) is repressed by H-NS, reviewed in [20], and CRP [21], but de-repressed by LeuO [22,23], and activated by addition to cells of *N^5^*, *N^10^*-methylene tetrahydrofolate, a product of bacterial glycine metabolism catalysed by enzymes of the Glycine Cleavage System [24]. The presence of the stress-responsive chaperone protein tHpG) is necessary for maintaining functional quantities of Cas3 in *E. coli* for CRISPR interference [25], and interaction with the Cascade-R-loop complex may constrain Cas3 activity until adaptation proteins (Cas1-Cas2) are also present [8], although this may not be the case in all Cascade-Cas3 containing systems. In *E. coli*, removal of H-NS (Δ*hns*) gives constitutive expression of Cas3 and Cascade that protects against λ phage plaques at 30 °C, but protection is lost at 37 °C [26]. The loss of CRISPR defence at 37 °C was reversed by ectopic overexpression of Cas3 protein [26]. It was known from the earliest studies of CRISPR systems that Cas3 is essential for defence against phage [5]—these latest genetic studies revealed a switch-like function for Cas3 that could be exposed by altering cell temperature. In this work, we investigated the mechanism of this property by analysis of Cas3 proteins, identifying key conserved tryptophan residues that are responsible by acting as a ‘gate’ to the passage of DNA into the nuclease active site. This provides experimental support for a structural analysis of Cas3 presented in [27], in which it was predicted that a tryptophan in this region of Cas3 may be important for protein activity.

## 2. Results

### 2.1. Cas3 Nuclease Activity Is Modulated by Temperature Changes That Correspond with Conformational Re-Arrangement of Cas3 in Transition from 30 °C to 37 °C

In CRISPR systems Cas3 nuclease is targeted to destruction of MGEs, and potentially provides MGE DNA as protospacers for updating the CRISPR immunity locus. In *E. coli*, we previously reported that the efficacy of CRISPR defence against phage λ is restricted by temperature [26,28]. This can be seen as increasing frequencies of phage λ plaques at 35–37 °C on lawns of Δ*hns E. coli* cells expressing Cas3, Cascade and λ-targeting crRNA when compared with plaques at 30 °C (Figure 1A). This effect was suppressed by ectopic (plasmid) over-expression of Cas3, compared to a control plasmid lacking *cas3* (Figure 1B). We investigated the mechanism for this effect using purified Cas3 proteins. *E. coli* wild-type Cas3 was purified fused to tandem (His)_6_- maltose binding protein (MBP) affinity tags, the latter to improve protein solubility and stability when expressed in *E. coli* [29,30]. It was clear from initial assays of Cas3 nuclease activity against a Cy5 end-labelled fork DNA (20 nM, Appendix A) that Cas3 (56 nM) was significantly more active at 30 °C compared to 37 °C (Figure 1C, compare panels i and ii). This DNA fork is a substrate for Cas3 nuclease targeting the 3’ ssDNA end. In denaturing gels (Figure 1C, panels iii and iv) this was apparent as removal of a few nucleotides at the 3’ end, giving the observed products because of the position of the 5’ Cy5 label—again nuclease activity was much reduced at 37 °C compared to 30 °C. This was consistent with the genetic data showing that *E. coli* is more susceptible to λ plaque formation at 37 °C because Cas3 is required for protection against them.

We investigated in more detail why Cas3 behaves in this temperature-sensitive way, reasoning that Cas3 may undergo temperature-induced conformational changes. Circular dichroism (CD) spectra were generated to detect changes in Cas3 secondary structure composition and thermodynamics when heated from 20 °C to 55 °C. This yielded thermograms of CD spectra changes at 222 nm, for alpha-helix and beta-strand composition [31], that, from three independent experiments, fitted to a Boltzmann equation gave a thermal inflection point at a mean of 34.75 °C with standard error +/−0.28 °C, summarized in Figure 1D with additional data in Appendix A. CD data were analyzed using the DichroWeb server [32], BeStSel [33] and CONTIN [34] to compare Cas3 protein secondary structure composition either side of the thermal inflection point at 34 °C, showing substantial change in α-helical and β-strand content (Table 1).

These data show temperature-dependent changes in Cas3 structure that correlate well with changes in nuclease activity and the function of Cas3 in CRISPR defence at 30 °C, which becomes inactivated at 37 °C (Figure 1A). We next used available crystal structures of Cas3 proteins to investigate in more detail the link between Cas3 nuclease activity, conformational movement and temperature.

### 2.2. A Single Tryptophan Residue (Trp-406) Modulates the Cas3 Nuclease Activity

Amino acid substitutions were made to Cas3 to identify the underlying factors that are responsible for the temperature dependent-nuclease activity and structural changes. The atomic resolution structure of *E. coli* Cas3 is not available, so we turned to the I-E subtype Cas3-DNA co-structures from *Thermobifida fusca* (PDB 4QQW-Z) [27] as a close homologue of the *E. coli* Cas3—30% amino acid identity, E-value 7e−54—to identify areas for mutagenesis. Many Cas3 proteins comprise a fusion of an HD-nuclease domain with superfamily 2 helicase (RecA-like) domains and an additional predicted C-terminal domain of unknown function (Figure 2A), reviewed recently in [18]. The interface between Cas3 HD and RecA1-like domains (iHDA1) includes a ssDNA binding channel in the vicinity of helicase motifs Ib and Ic, which interacts with ssDNA in superfamily 2 helicases, that leads to the HD nuclease active site (Figure 2A,B). The iHDA1 has been predicted to have conformational flexibility for reorientation of ssDNA when gaining access to the nuclease active site [27,35]. We targeted two tryptophan residues that are highly conserved across bacterial Cas3 proteins (Figure 2C), one located in the iHDA1 (Trp-406 in *E. coli* Cas3 numbering) and the other (Trp-230) located close by (Figure 2A,B), and both of which are predicted to interact with ssDNA [27].

Mutant Cas3 proteins were purified with amino acid substitutions in Trp-230 or Trp-406, and in two further residues, Trp-149 and Trp-152, that, although located close to Trp-230 and Trp-406 (Figure 2B), are much less well conserved across species and were used in comparison with Trp-230/406. We therefore purified mutant proteins Cas3^W230A^, Cas3^W406A^, Cas3^W149A^ and Cas3^W152A^ for assays measuring their nuclease activity compared to the wild type protein. Using the DNA fork (20 nM) substrate and measured as a function of time, wild type Cas3 (56 nM) converted maximally 92% of DNA into nuclease product at 30 °C, compared to 40% at 37 °C (Figure 2D). Cas3^W149A^, Cas3^W152A^ and Cas3^W230A^ (all at 56 nM) were significantly less active than wild type Cas3 at both temperatures. However, Cas3^W406A^ (56 nM) was active similarly to wild type Cas3 at 30 °C and hyperactive compared to wild type Cas3 at 37 °C (mean of 73% fork converted to nuclease product). Therefore, the Cas3 Trp-406 residue seemed to modulate Cas3 nuclease activity because changing it to alanine substantially overcame the inhibitory effect of elevated temperature on nuclease activity. There was no evidence in Electrophoretic Mobility Shift Assays (EMSAs) for Cas3^W406A^ binding to DNA more stably or with greater efficacy than wild type Cas3 (Appendix A panels (i and (ii), although the mutant proteins with reduced nuclease activity (Cas3^W149A^, Cas3^W152A^ and Cas3^W230A^) formed less pronounced protein-fork DNA complex in the same EMSAs (Appendix A panels iii–vi).

### 2.3. Evidence That Trp-406 Is a ‘Gate’ in Cas3 That Controls Access of ssDNA to the Nuclease Active Site

When *E. coli* Cas3^W406A^ was assessed using CD and BeStSel in the same way as for wild type Cas3, we observed the thermal melting/inflection point increased from 34.75 °C to 36.7 °C (Appendix A), accompanied by changes in β-strand that were significantly different from those observed for wild type Cas3 (Table 2). We next used molecular dynamics simulations focusing on Trp-406 to investigate why the Cas3^W406A^ protein may behave in this way.

In the Cas3 crystal structure from *T. fusca* (PDB 4QQW) [27], the tryptophan equivalent to *E. coli* Cas3 Trp-406 is abutted to another tryptophan residue (Trp-230 in *E. coli*). This Trp-230/406 pair is close to the ssDNA binding channel in the iHDA1 (Figure 2B,C). We used this structure and that of Cas3 from *Thermobaculum terrenum* (PDB 4Q2C) [35] to generate via SWISS-MODEL [36] a predicted structure for *E. coli* Cas3 to begin molecular dynamics simulations of these tryptophan residues at 30 °C and at 37 °C. In molecular dynamics simulations, we identified temperature dependence for interaction of the highly conserved residues Trp-406 and Trp-230. Increasing temperature resulted in increased distance, from a mean of approximately 6.1 Å at 30 °C to 11.6 Å at 37 °C (Figure 3A and Appendix A)**.** When in close proximity (30 °C), Trp-230 and Trp-406 from a stable π–π interaction that was broken in the first phase of MD simulations at 37 °C and remained broken through the rest of simulations (Figure 3B). The interaction breaks because of Trp-406 moving away from Trp-230 to a position close to the Ic helicase motif (Asn-Lys-Lys, see also Figure 2A) comprising residues 411–413. There, Trp-406 is stabilized into a new position by hydrophobic interactions with newly surrounding residues (Figure 3B, right panel, Appendix A). At 30 °C in MD simulations, Trp-406 bound to Trp-230 lies beside the ssDNA binding tunnel that connects the translocation motifs with the HD nuclease active site (Figure 3C panel i). However, at 37 °C, positioning of Trp-406 forms an obstruction within the tunnel that would limit ssDNA access to the nuclease active site (Figure 3C panel ii)**.** Therefore, Trp-406 is acting as a gate to control access of ssDNA to the nuclease active site. This provides a molecular explanation for lower activity of Cas3 as a nuclease at 37 °C compared to 30 °C, and why Cas3^W406A^ is hyperactive as a nuclease at 37 °C—replacement of Trp-406 with alanine would have the effect of leaving the gate open for passage of ssDNA.

### 2.4. Hyperactivity of Cas3^W406A^ Nuclease Is Independent of Cas3 ATPase Activity and of the Cascade Complex

CRISPR interference requires Cas3 nuclease and ATP-dependent DNA translocase activities [5,37]. Structural models of Cas3 function describe ATPase activity that powers protein conformational movement that results in translocation of ssDNA into the HD nuclease active site, via the ssDNA binding channel through the helicase domains. As noted above (Figure 2C), Trp-406 is located close to the nuclease active site and the ssDNA channel, at the interface with domain RecA1 that contains the ATPase active sites. We therefore tested if Cas3^W406A^ was also a hyperactive ATPase that, based on the structural models, may explain hyperactive nuclease activity. Cas3 ATP hydrolysis was measured using the malachite green reporter assay [38] at 30 °C and 37 °C, compared to reactions lacking Cas3. In contrast to nuclease activity, temperature had no significant effect on wild type Cas3 ATPase activity (Figure 4A)**.** ATPase activity of each Cas3 mutant was similar whether measured at 30 °C or 37 °C, also contrasting with the much-reduced nuclease activity from three of the mutants at the higher temperature. Cas3^W406A^ was not hyperactive as an ATPase at 37 °C. Therefore, the effect of temperature and the Trp-406 residue are exerted only on Cas3 nuclease activity.

In *E. coli* CRISPR interference, Cas3 is recruited to MGE DNA at an R-loop formed by the Cascade ribonucleoprotein complex [16]. Cas3 then degrades the non-targeted MGE DNA strand [8,19,39,40,41,42]. The Cas3 nuclease reaction is in two steps, first, magnesium-dependent nicking of supercoiled DNA [8,40], and then ATP-dependent translocation of ssDNA to power processive nucleolytic digestion [37,43]. The interplay of Cas3 and Cascade in interference reactions led us to test for Cas3 and Cas3^W406A^ nuclease activities against a plasmid targeted by Cascade in vitro (Figure 4B). The Cascade apo-protein complex was purified and activated by binding to a targeting CRISPR RNA (crRNA)—the complex targets M13 double-stranded supercoiled DNA to form an R-loop, seen as a slower migrating M13 species compared to reactions lacking Cascade (Figure 4B, compare lanes 2 and 3) [44]. Cas3 converted supercoiled M13 nto nicked and linearized DNA in reactions dependent on magnesium (compare lanes 4 and 5), and the same effect was observed using Cas3^W406A^ instead of wild type Cas3 (lane 6)**.** An interesting result was observed when Cascade R-loops were pre-formed on M13 DNA—in these reactions, wild type Cas3 nuclease activity was inhibited by Cascade, observed as the absence of linearized M13 and re-appearance of supercoiled M13 (Figure 4B lane 7), but activity of Cas3^W406A^ in this respect was unaffected by Cascade (lane 8).

Assays comparing Cas3 and Cas3^W406A^ nuclease activity in the presence of Cascade were repeated with the addition of ATP, to observe for processive nucleolytic degradation of the M13 DNA (Figure 4C)**.** Consistent with a previous analysis of CRISPR interference in vitro [29], reactions containing Cascade, wild type Cas3 and Mg-ATP gave an accumulation of nicked DNA (Figure 4C, compare lanes 3–5 with lanes 1 and 2). In contrast, though, Cas3^W406A^ generated products consistent with processive nuclease activity (Lane 6), uncoupled from the constraint of Cascade. This intriguing observation indicates that the nuclease hyperactivity of Cas3^W406A^ can also overcome the modulating effect of Cascade in vitro.

### 2.5. Cas3^W406A^ Does Not Protect E. coli Cells from Phage Lysis

Finally, we evaluated if Cas3^W406A^ nuclease hyperactivity impacted CRISPR catalyzed protection against phage lambda (λ*vir*). The assay requires expression of Cas3 and Cascade in *E. coli* to determine if these cells are resistant to lambda plaque formation [5,30]. The CRISPR−1 locus of these cells was engineered to contain two anti-λ*vir* spacers for targeting of Cascade to λ*vir* DNA. In addition, the native *ygcB* gene encoding wild type Cas3 within the CRISPR-cas locus was replaced by the *ygcB*^W406A^ allele, encoding Cas3^W406A^. Cells were lacking H-NS (Δ*hns*) to de-repress transcription of Cascade and Cas3 encoding genes. At 30 °C, wild type Cas3 cells were fully protected from plaque formation when challenged by λ*vir*, but this protection was lost at 37 °C (Figure 5), as expected from Figure 1 and [28]. Cas3^W406A^ cells showed λ*vir* plaques at both temperatures, and, in addition, Cas3^W406A^ plaques were significantly larger than those of wild type Cas3 at 37 °C, further indicating that the Cas3^W406A^ mutation is ineffective for CRIPSR interference (Figure 5). We observed no difference between the morphology or viability of unchallenged Cas3 or Cas3^W406A^ expressing cells that might explain the difference. Therefore, we conclude that, although Cas3^W406A^ is a hyperactive nuclease, its expression in cells is unable to deliver the CRISPR effector response that protects against phages. This interesting phenomenon is discussed below because it may highlight allosteric regulation of Cas3 in cells.

## 3. Discussion

In this work, we found that temperature sensitivity of Cas3 nuclease in vitro is independent of ATPase activity, with evidence that this is caused by positioning of a highly conserved pair of tryptophan residues close to the nuclease active site. The in vitro data is in agreement with previous genetic analysis identifying temperature sensitivity of Cas3 in cellular defence against phage λ, which we can now attribute to the loss of Cas3 nuclease function. The tryptophan residues (Trp-230/406) reside in a region of Cas3 we refer to as iHDA1, a domain interface needed to channel ssDNA between the RecA1 helicase domain and the HD nuclease active site. In a structural analysis of Cas3 [27], the DNA phosphate backbone is described as undergoing extraordinary twisting in the iHDA1 in order to reach the HD active sites. Our analyses show that the position of Trp-406 in *E. coli* Cas3 may be either ‘permissive’ or ‘inhibitory’ to ssDNA accessing the nuclease active site (Figure 6), and that this alone may explain the temperature-sensitive behaviour of Cas3. CD spectroscopy showed temperature-dependent changes in Cas3 secondary structure, including a major effect at 34.75 °C consistent with a switch in protein conformation, and CD spectra changes associated with mutating Trp-406 to alanine. In molecular dynamics simulations, we observed that the Trp-406 R-group moves from lining the ssDNA binding channel of iHDA1 at 30 °C (DNA permissive), to forming a steric block to the channel at 37 °C (inhibitory). Inhibition of nuclease activity at 37 °C was substantially overcome by replacing Trp-406 with alanine. In effect, the Cas3^W406A^ mutation has generated a hyperactive Cas3 nuclease. However, this mutation was disadvantageous to cells for CRISPR defence against phage because replacement of the wild type *cas3* allele on the *E. coli* chromosome with *cas3*^W406A^ made cells susceptible to phage lysis, seen as plaques on bacterial lawns. The invariance of Trp-406 across bacterial Cas3 proteins is consistent with it holding a crucial role for Cas3 function in CRISPR systems, which had not been previously identified, although a structural study of Cas3 had highlighted this tryptophan as likely to be important in Cas3 function [27].

Cas3 structure therefore exerts intrinsic control over nuclease activity, via a gate formed of Trp-406 opening and closing access of ssDNA to the nuclease active site (Figure 6).

This tryptophan residue, and its binding partner at lower temperature, Trp-230, is well positioned for this, being close to the HD active site—a 12 Å mean distance from the centre of mass of the Trp-406/Trp-230 pair and the HD active sites in MD simulations at 30 °C. Allosteric regulation of nucleases in biological systems ensures that they are targeted to destroy only appropriate substrate nucleic acids at an appropriate time. For Cas3, the interaction with Cascade during CRISPR interference provides targeting, and functional levels of Cas3 in cells rely on the presence of the chaperone HtpG [25], another form of post translational control over Cas3. In a speculative model, we suggest that HtpG may chaperone Cas3 via physical interaction that suppresses Cas3 nuclease activity until it encounters the Cascade complex, with which it interacts physically through the Cse1/CasA subunit. In this scenario, exchange of Cas3 from HtpG to Cascade is accompanied by changes to the iHDA1 region tryptophan residues that allow ssDNA to be degraded, equivalent to the effect we observe to be brought about by temperature changing from 37 °C to 30 °C. Therefore, the effect of temperature that we observe may be a proxy for changes induced by HtpG. However, growth temperature and other environmental factors have a profound effect on bacterial cell physiology [45], and therefore the effect of temperature that we observe on Cas3 function may reflect environmental control of CRISPR systems. Other factors could instead be involved, including post-translational chemical modifications to Cas3 protein [46], although the known requirement for HtpG and Cas3 to co-function in CRISPR immunity would make such an effect of HtpG the most likely explanation at present. Further in vitro studies of Cas3 alongside HtpG, and Cascade R-loops, should enable these ideas to be tested.

Bacterial Cas3 and Cascade have been deployed for genetic insertions or deletions in human cells [47,48,49]. These reactions rely on the processive nuclease activity of Cas3 to delete up to 200 kb of DNA from the human cell chromosome targeted by the Cascade complex, with potentially therapeutic uses [50]. The identification of hyperactive nuclease Cas3^W406A^ may be of use in this respect, particularly if it can be successfully targeted by Cascade but then has nuclease activity that is not modulated by Cascade.

A concluding summary of the work presented here is shown in Figure 6.

## 4. Materials and Methods

### 4.1. E. coli Strains, Plasmids and Molecular Cloning

*E. coli* strains and their derivation are listed in Appendix A. Plasmids and primers are listed in Appendix A. The gene encoding Cas3 (*ygcB*,) was amplified from *E. coli* MG1655 genomic DNA and cloned into pBAD-HisA (Thermo Fisher Scientific, U.K.) *via Xho*I and *Eco*RI sites to construct plasmid pAH4 for overexpression and purification of Cas3 with an N-terminal histidine tag. The plasmid for purification of (His)_6_-MBP-Cas3 protein was described in [29]. Point mutants used in this study were created using a Q5 Site-Direct Mutagenesis Kit (New England Biolabs) and plasmids were verified by Sanger sequencing (Source Bioscience).

### 4.2. Protein Expression and Purification

Plasmid encoding *ygcB* with either a (His)_6_-MBP tag (pCas3) or (His)_6_-tag only (pAH4) was transformed into BL21 A.I. (Invitrogen) and selected on agar containing 100 µg/mL ampicillin. Cells were grown in LB broth with 100 μg/mL ampicillin at 37 °C until the absorbance at 600 nm (OD_600_) reached 0.3 and L-arabinose 0.2% (*w/v*) was added at this point to induce protein overexpression. Growth continued for another 3 h at 37 °C before cells were resuspended in lysis buffer (20 mM Tris-HCl pH 8.0, 100 mM NaCl, 10% (*v/v*) glycerol and 0.5 mM phenylmethylsulphonyl fluoride (PMSF)) and stored at −80 °C. Cells were lysed and centrifuged at a relative centrifugal force of 48,000 g for 1 h at 4 °C.

Cell lysates containing over-expressed Histidine-tagged Cas3 were loaded onto Ni-NTA resin pre-equilibrated with lysis buffer, and bound proteins washed with a further buffer (50 mM Na_3_PO_4_ buffer pH 8.0, 300 mM NaCl, 20 mM imidazole). Proteins were eluted from the resin via a linear gradient of increasing imidazole (50 mM to 125 mM) in 50 mM Na_3_PO_4_ buffer pH 8.0 and 300 mM NaCl. Samples were analyzed by 10% (*w/v*) SDS-PAGE and Western blotting with anti-histidine antibody (Monoclonal Anti-polyhistidine-Peroxidase clone HIS-1, Sigma-Aldrich). Proteins were concentrated with ultrafiltration in an Amicon Ultra 3K device (Merck) in 50 mM Na_3_PO_4_ buffer pH 8.0 and stored at −80 °C.

Cell lysates containing over-expressed (His)_6_-MBP-Cas3 were loaded onto Ni-NTA resin (GE Healthcare) pre-equilibrated with buffer A (20 mM Tris-HCl pH 8.0, 10 mM imidazole, 100 mM NaCl, 10% (*v/v*) glycerol). The column was then washed with a further buffer (20 mM Tris-HCl pH 8.0, 10 mM Imidazole, 1 M NaCl, 10% (*v/v*) glycerol) and re-equilibrated with buffer A. (His)_6_-MBP-Cas3 was directly eluted from Ni-NTA resin with buffer B (20 mM Tris-HCl pH 8.0, 250 mM Imidazole, 100 mM NaCl, 10% (*v/v*) glycerol), and recovered proteins were loaded into 1 mL heparin column (GE Healthcare, U.K.) and then 1 mL MBPTrap HP column (GE Healthcare) both pre-equilibrated with buffer MA (20 mM Tris-HCl pH 8.0, 100 mM NaCl). Cas3 did not bind to heparin column and therefore the column flow through was loaded directly into the MBPTrap HP column. Proteins were eluted from the MBPTrap HP column with buffer MB (20 mM Tris-HCl pH 8.0, 100 mM NaCl, 10 mM maltose) and dialysed into buffer containing 20 mM Tris-HCl pH 8.0, 100 mM NaCl and 30% (*v/v*) glycerol for storage at −80 °C.

### 4.3. DNA Substrate Preparation

The DNA fork substrate for evaluating Cas3 nuclease activity shown in Appendix A was constructed by annealing 5 μM of each oligonucleotide MW12 and MW14 in buffer comprising 10 mM Tris pH 7.5, 50 mM NaCl and 1 mM EDTA. The DNA was incubated at 95 °C for 10 min allowed to cool to 20 °C overnight. The annealed DNA fork was separated from free oligonucleotides by migrating the sample on a 10% acrylamide 1 x TBE gel. The band containing the fork substrate was excised, with annealed fork substrate eluted in elution buffer (4 mM Tris pH 8.0 and 10 mM NaCl), followed by ethanol precipitation to concentrate the fork substrate.

### 4.4. Protein-Nucleic Acid Assays

Cas3 proteins were incubated with 20 nM of fork DNA in buffer O (50 mM Tris-HCl pH 7.5, 10 mM MgCl_2_, 100 mM NaCl, 0.1 mg/mL BSA, 20 mM DTT), at 30 or 37 °C. Reactions were stopped by adding 50 mM Tris pH 8.0, 100 mM EDTA, 5 mg/mL proteinase K and 1% (*w/v*) SDS, followed by incubation at 37 °C for 15 min. Reactions were mixed with Orange G loading dye 1 (80% (*v/v*) glycerol, Orange G) for electrophoresis in 10% acrylamide 1 x TBE gels, or Orange G loading dye 2 (20% *v/v* glycerol, 78% *v/v* formamide and Orange G) for electrophoresis through 10% acrylamide denaturing gels.

For EMSAs, Cas3 proteins were incubated with 20 nM of fork DNA in buffer E (50 mM Tris-HCl pH 7.5, 100 mM NaCl, 0.1 mg/mL BSA, 20 mM DTT), at room temperature for 1 h. Reactions were mixed with 8 μL Orange G loading dye 1 (80% *v/v* glycerol and Orange G) for electrophoresis in 10% acrylamide 1 x TB gel containing 0.13 M Tris-HCl (pH 7.6), 45 mM boric acid, 8% acrylamide and 5 mM DTT. Running buffer containing 0.13 M Tris-HCl (pH 7.6), 45 mM boric acid, 5 mM DTT, and 0.1 mg/mL BSA was used in EMSA.

For Cas3 nuclease reactions in the presence of Cascade, the Cascade complex was prepared as described in [44]. The crRNA sequence, given in the Appendix A, anneals to M13 RF I DNA. To analyse Cas3 nicking activity, 100 nM Cascade was first pre-incubated with 50 ng of M13 RF I (supercoiled) DNA in buffer O (50 mM Tris-HCl pH 7.5, 10 mM MgCl_2_, 100 mM NaCl, 0.1 mg/mL BSA, 20 mM DTT) at 30 or 37 °C for 30 min, to form an R-loop complex. Cas3 proteins were added and the mixture incubated for a further 120 min before directly mixing with Orange G loading dye (80% *v/v* glycerol and Orange G) and loading for electrophoresis through a 1% TAE agarose gel. If ATP was to be included, it was added to a final concentration of 2 mM in the Cas3 protein sample. All reactions were stopped using 50 mM Tris pH 8.0, 100 mM EDTA, 5 mg/mL proteinase K and 1% SDS, before adding Orange G dye for electrophoresis in a 1% TAE agarose gel.

### 4.5. ATPase Assays

Cas3 ATPase assays followed the method described in [38] that uses malachite green dye as a reporter for liberation of phosphate from ATP. Reactions were in buffer O (50 mM Tris-HCl pH 7.5, 10 mM MgCl_2_, 100 mM NaCl, 0.1 mg/mL BSA, 20 mM DTT) supplemented with 20 nM of MW14 ssDNA and 2.5 mM ATP. Reactions were incubated at either 30 °C or 37 °C for 60 min before adding eight reaction volumes of pre-mixed colour reagent, incubating at room temperature for 2 min and stopping the reaction by adding one volume of 3% (*w/v*) sodium citrate. Colour reagent was prepared using six volumes of 0.045% (*w/v*) malachite green hydrochloride mixed with one volume of 4.2% (*w/v*) ammonium molybdate in 4 M HCl. Samples were transferred to 96-well flat bottom plate (Life Technologies) to measure phosphate production by dye absorbance at 660 nM. The total amount of phosphate product (nmol) was calculated from a phosphate standard curve plotted using NaH_2_PO_4_ at 2–16 nmol in 100 μL reactions developed as described above.

### 4.6. Phage Sensitivity Assay by Plaque Formation

This assay was done according to reference [26]. Cells were grown to saturation overnight in LB medium at 37 °C with 0.2% (*w/v*) maltose. LB plates were overlaid with 3 mL of 0.6% (*w/v*) LB top agar containing 0.2 mL of cells. Then, 10 μL aliquots of serially diluted λ*vir* phage in 10 mM MgSO_4_ were spotted on the surface of set agar and allowed to soak in. Plates were incubated overnight at 30 °C or 37 °C. The sensitivity of the cells to infection was represented as the plaque-forming units (PFUs) by plaques from several dilutions being counted and their number per ml calculated.

### 4.7. Construction of the Chromosomal Cas3 Point Mutant

The *cas3*^W406A^ allele was made in pAH4 by site-directed mutagenesis to create plasmid pIIB39—details of the primers and plasmids are in the Appendix A. This was used to replace the wild type Cas3 allele in *E. coli* strain IIB1309 (T3 λc Δc*as1*::FRT Δ*hns*::*kan*) using the gene replacement method described in [51]. The full Cas3 encoding gene (*ygcB*) containing the mutation for W406A was cloned from plasmid pIIB39 into the pKOV plasmid using the NEBuilder^®^ HiFi DNA Assembly kit and primers listed in Table 2 (oligos). The assembly reaction was prepared according to the manufacturer’s recommendations and 2 µL of the mixture was transformed into NEB 5-alpha competent *E. coli* cells and incubated over night at 30 °C. Transformed colonies were screened using PCR and sequencing for the presence of the correct *cas3* gene insert. Correct plasmid with the insert was purified and transformed into the strain IIB1309 (λT3 λc Δ*cas1*::FRT Δ*hns*::kan) and plated onto chloramphenicol agar to induce plasmid integration at 42 °C. From the 42 ° C plate, 1–2 colonies were picked into 3 mL of LB broth with 5% *w/v* sucrose, grown over night at 37 °C and plated next day onto LB plate with 5% *w/v* sucrose to induce loss of the replacement vector. Individual colonies were then tested by phage infection assay and loss of antibiotic resistance. Colonies that showed plaques at 30 °C and sensitivity to antibiotic chloramphenicol were selected. The *cas3* gene was amplified and sequenced in the Macrogen Europe service to confirm the point mutation.

### 4.8. Circular Dichroism Measurements and Analysis of Cas3 Secondary Structure

Circular dichroism (CD) measurements of purified (His)_6_-Cas3 used a Jasco J-815 CD spectrophotometer connected to a Peltier temperature control system. Protein samples were diluted in 50 mM sodium phosphate (pH 8.0) to a concentration of 0.1–0.2 mg/mL in a total volume of 300 µL, for analysis in a 1 mm path length thermo-cuvette. Spectra were measured from 190 nm to 260 nm, and at 222 nm, at 20 °C with continuous scanning (50 nm/min) using a response time of 1 s and bandwidth of 1 nm. Delay was set at 30 s, sensitivity as standard and data pitch as 0.2 nm. For an assessment of the thermal denaturation of Cas3 proteins, the CD ellipticity (mdeg) was recorded at 222 nm in the temperature range from 20 °C to 55 °C for Cas3 and from 30 °C to 90 °C for archaeal Cas3 with a temperature slope of 1 °C/min and data pitch set as 1 °C or 5 °C, with delay of 30 s for each temperature. The buffer CD signal was subtracted from all measurements. Data were collected considering the HT voltage applied to detector was at the maximum of 600 V. Data were processed and fitted to a Boltzmann sigmoidal curve using the Origin software. Changes in the secondary structures of the Cas3 and Cas3^W406A^ proteins were calculated at two temperatures using the CD experimental data at Dichroweb [32], BeStSel [33] and algorithm CONTIN with set 4 parameters (optimized for 190–240 nm) [32,34].

### 4.9. Molecular Dynamics Simulations of Cas3 Protein

Starting models of *Escherichia coli* Cas3 were prepared from structures of Cas3 from *Thermobaculum terrenum* (PDB ID: 4Q2C) [35] and *Thermobifida fusca* (PDB ID: 4QQW) [27] using the SWISS-MODEL web server [36] (Appendix A). Polar hydrogen atoms were added using H++ [52] and WHATIF software [53] in order to obtain an optimal hydrogen bonding network. Non-polar hydrogen atoms were added using the *tleap* module of Amber16 program package [54]. Protein structures were placed in the centre of the box filled with a TIP3P model of water molecules. Parameters for the protein atoms, Mg^2+^ ions and DNA atoms were obtained from the Amber ff14SB and the Amber OL15 force fields. Parameters for ATP were obtained from literature. Before molecular dynamics (MD) simulations, systems were energy minimised in several cycles using different constraints (the detailed procedure is described in [55]. During the first 250 picoseconds (ps) of MD simulations, temperature was linearly increased from 0 K to final temperature (301 K, 303 K or 310 K, depending on the simulation), while, from 250 ps to 300 ps, temperature was kept constant. During the first 300 ps, volume was kept constant, and solute atoms were constrained using a harmonic potential with a force constant of 32 kcal/(mol·Å^2^). For the next 200 ps, the system was equilibrated at a constant temperature under constant pressure using the weak coupling Berendsen thermostat algorithm with the time constant for heat bath coupling set to 2 ps. During the production phase, systems were simulated at constant pressure with a Berendsen thermostat [56] and the time constant for heat bath coupling set to 1 ps. The time step of the simulation was 1 fs and structures were sampled every 100 ps. Periodic boundary conditions (PBC) were applied and electrostatic interactions were calculated using a particle mesh Ewald; systems were made electroneutral by addition of Na^+^ ions. Simulations were carried out using the Amber 16 and simulation packages. The following systems were subjected to computational simulations: (i) protein complexes with two Mg^2+^ ions and ATP, (ii) protein (wild type and mutants) complexes with two Mg^2+^ ions and ssDNA. Each system was simulated at lower (301 K and/or 303 K) and higher (310 K and/or 317) temperature (Appendix A and Appendix A). The trajectories were analysed using VMD [57], UCSF Chimera [58], the Cpptraj program from the Amber 16 package [54] and the MOLE 2.5 program [59]. Data were analysed and plotted using the program package R with the help of libraries ggplot2, data.table and stringr.

## Figures and Tables

**Figure 1 ijms-22-02848-f001:**
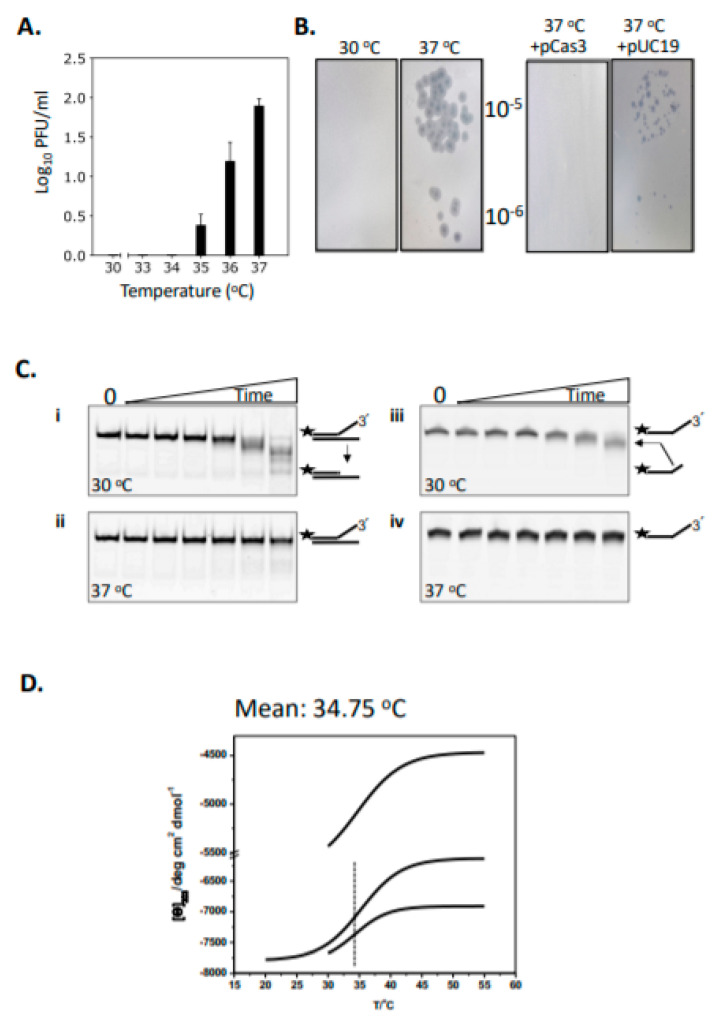
Temperature-induced changes to Cas3 structure and function in *E. coli* cells and in vitro. (**A**) Δ*vir* plaque formation was measured at 30-37 °C as a read-out for the efficacy of CRISPR interference. Plaques (PFU, plaque forming units) were readily observed at and above 35 °C (see also part **B**). Assays were in triplicate and standard errors from the mean are shown. (**B**) (i) Plaques formed as measured in part A at 37 °C were not observed if the cells also over-expressed Cas3 from a plasmid (ii, +pCas3), compared with the empty plasmid vector (pUC19) used as a control (iii). (**C**) Cas3 nuclease activity degrades Cy5 end-labelled (25 nM, label denoted by a star) DNA much more effectively at 30 °C compared to 37 °C, in reactions containing 56 nM of Cas3 with samples taken at 0, 5, 15, 30, 60, 120 and 240 min. Panels i and ii show native TBE polyacrylamide gels, and iii and iv are denaturing (urea) TBE gels. (**D**) Shows the Boltzmann curve traces from each of three independent experiments for *E. coli* Cas3, derived from circular dichroism spectroscopy measurements at 222 nm. The mean temperature for the inflection point is shown—34.75 °C—and see Appendix A for the full plotted data.

**Figure 2 ijms-22-02848-f002:**
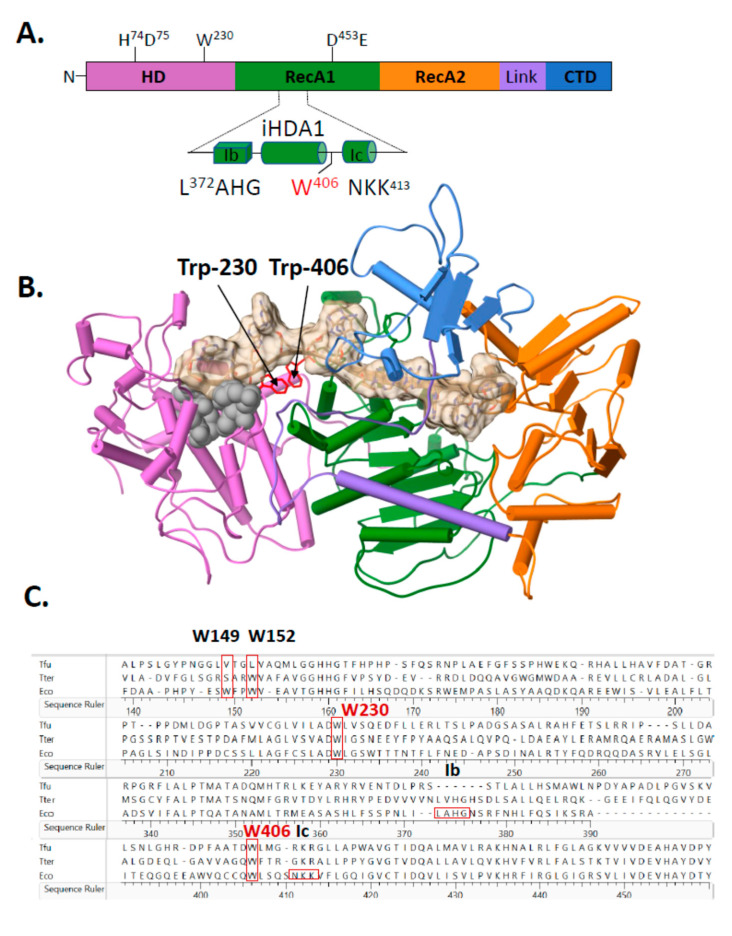
Structural positioning of two invariant tryptophan residues at the interface of Cas3 HD and RecA1 domains—a key role in Cas3 nuclease function for Trp-406. (**A**) Cartoon representation of Cas3 protein with amino acid residues indicated for the nuclease (HD) and Walker B ATPase active sites using numbering from the *E. coli* protein. Highlighted in the foreground is the region detailed in the results, located at the interface of HD and RecA1 domains (‘iHDA1’) and including the ssDNA binding helicase motifs Ib and Ic. (**B**) *E. coli* Cas3 structure deduced from T. fusca Cas3 (PDB: 4QQW) and T. terrenum (PDB: 4Q2C) (27,35) highlighting the tryptophan residue W406 (labelled) and the passage of ssDNA (tan cord). Cas3 regions are denoted as follows in the same way as in part A: HD (orchid), RecA1 (green), RecA2 (orange), an interdomain linker helix (purple) and the CTD (blue). Grey spheres indicate the HD nuclease active site residues. (**C**) Sequence alignment highlighting conservation of residues Trp-406 and Trp-230 in Escherichia coli K-12 (Eco) with the structurally determined Cas3 proteins from T. fusca (Tfu) and T. terrenum (Tter) proteins. Protein sequences were aligned using Clustal Omega and results were exported via Lasergene 17. (**D**) Nuclease activity of Cas3 and mutants (56 nM) measured on the DNA fork substrate (20 nM). Samples were collected at 0, 5, 15, 30, 60, 120 and 240 min. Reactions were carried out in triplicate at 30 °C or 37 °C as indicated, and data points show standard errors from the mean.

**Figure 3 ijms-22-02848-f003:**
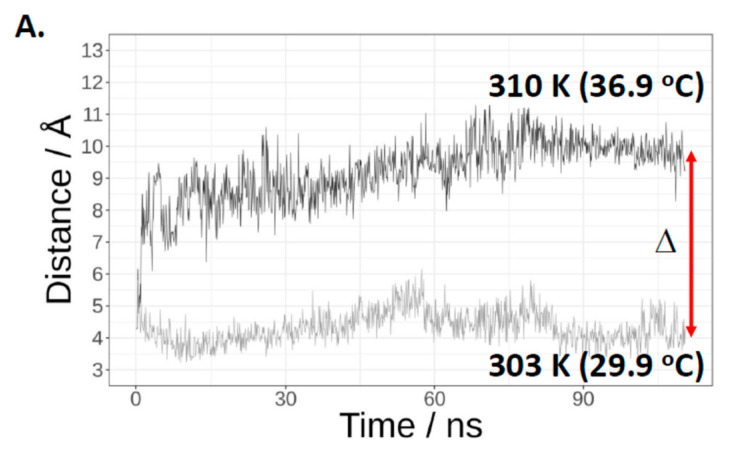
Molecular dynamics simulations indicate significant temperature-induced changes in positioning of Trp-230 and Trp-406. (**A**) Shows the distances between the centre of phenyl rings of Trp-230 and Trp-406 monitored during the MD simulations at 30 °C and 37 °C, as indicated. (**B**) Structures obtained during the final phase of MD simulations. The panel left side shows the Trp-230 and Trp-406 π-π interaction at 30 °C that is disrupted at 37 °C (panel right) because Trp-406 is moved into the pocket where it is stabilized by hydrophobic interactions within the ssDNA tunnel, shown as surfaces representing Thr-392, Gln-402, Gln-405, Leu-407, Gln-409, Lys-412, Lys-435, His-436, Asp-780 and Asp-778. (**C**) Temperature-induced changes in the positioning of Trp-230 and Trp-406 result in Trp-406 movement and blockage of the ssDNA binding channel (shown as grey surfaces representing the residues that form it), which we propose explains the lack of nuclease activity shown by *E. coli* Cas3 at the higher temperature, and its alleviation by replacing tryptophan with alanine. Cas3 protein is coloured in Cyan and Trp-230 and Trp-406 are coloured in red. At 37 °C Trp-406 moves to a location within a pocket in the ssDNA tunnel.

**Figure 4 ijms-22-02848-f004:**
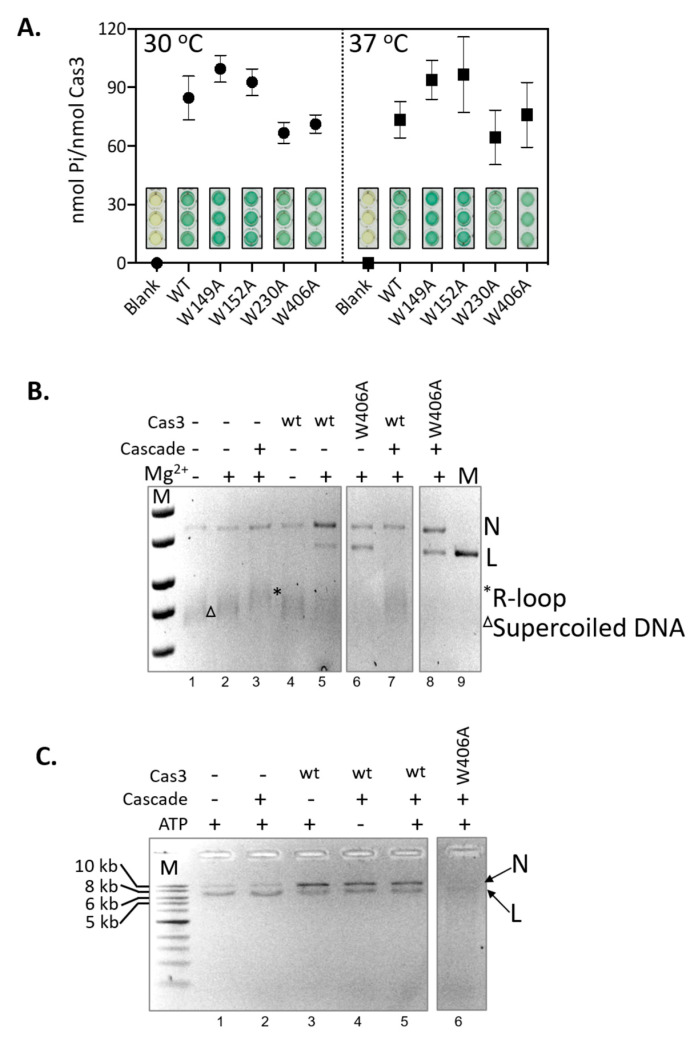
Nuclease hyperactivity of Cas3^W406A^ is independent of ATPase activity and interaction with Cascade. (**A**) End-point measurements of ATP hydrolysis by Cas3 proteins (790 nM) was compared at 30 and 37 °C as indicated. Reactions were in triplicate, standard errors from the mean are shown, to give measurements of phosphate (Pi) released from *per* nM of Cas3 protein. These values were obtained from spectroscopic measurements of malachite green dye intensity, shown in the graph, each measurement was obtained after blanking the spectrophotometer with reactions lacking Cas3. (**B**) Shows Mg^2+^-dependent DNA nicking activity of Cas3 and Cas3^W406A^ (850 nM) on M13 DNA in reactions that contain preformed Cascade (100 nM) R-loop substrates, indicated by an *. Reaction components are as indicated above the ethidium bromide stained agarose gel panel. In these reactions no deproteinising ‘stop’ solution was added, so that we were able to visualise that Cascade R-loops had formed (e.g., lane 3). (**C**) Shows Cas3 nicking and nuclease activity in the presence of Mg^2+^ and ATP and Cascade (100 nM), as indicated above the ethidium bromide stained agarose gel panel. These reactions were stopped by adding proteinase K ‘stop’ solution to dissipate R-loops so that nuclease products are more easily discerned. For panels (**B,C**) the topology of the DNA is highlighted to the right of the gel image by N (nicked) L (linear), * (R-Loop) and Δ (Supercoiled).

**Figure 5 ijms-22-02848-f005:**
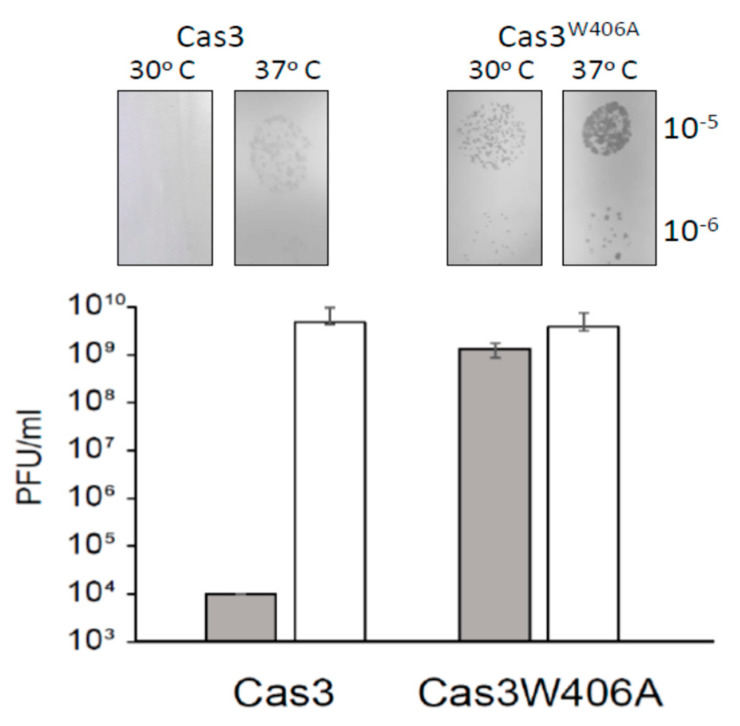
The allele *cas3*^W406A^ does not provide resistance to λ*vir* at 30°C. *E. coli* cell lawns of strain Δ*hns* (Δ*cas1*) + λc + λT3 (IIB1309) and *cas3*^W406A^ Δ*hns* (Δ*cas1*) + λc + λT3 (IIB1342) were infected with phage dilutions (from 10^−3^ to 10^−7^) and incubated at 30°C and 37°C as indicated. Bars represent average and SD of the number of plaque forming units (PFUs) per ml from three independent experiments. Also shown are corresponding photographs of typical bacterial lawns and viral plaques giving the data.

**Figure 6 ijms-22-02848-f006:**
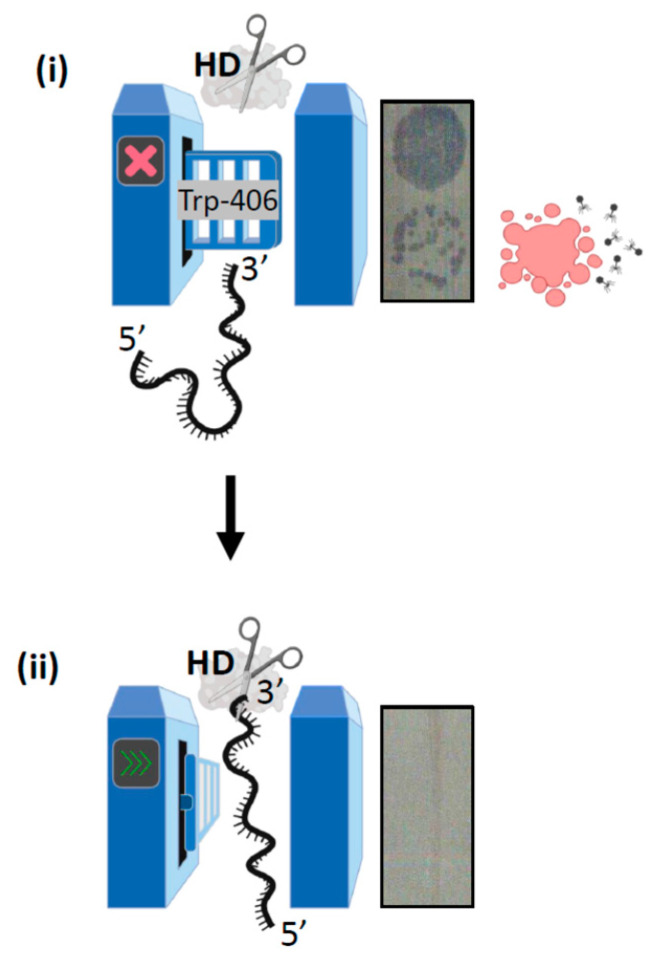
Concluding summary of the Trp-406 nuclease ‘gate’ model. (**i**) When Trp-406 is positioned close to helicase motif Ic it blocks the ssDNA binding channel to the nuclease (HD) active site, resulting in a ‘closed’ gate. In this scenario CRISPR interference is inhibited because MGE DNA is not destroyed, resulting in cell lysis evident as plaques on a bacterial lawn. (**ii**) Conformational movement of Cas3 re-positions Trp-406 proximally to Trp-230, with which it interacts to form a lining to ssDNA channel in the gate being ‘open’. In this state Cas3 is able to degrade MGE DNA to fulfil its role in CRISPR adaptation reactions.

**Table 1 ijms-22-02848-t001:** Cas3 secondary structure from CD measurements at either side of the inflection point.^a.^

Secondary Structure	% Composition in Cas3 at 30 °C	% Composition in Cas3 at 35 °C	Change (Δ)
α-Helix	18.1	14.3	−3.8
β-antiparallel	11.7	14.9	3.2
β-parallel	0.0	0.0	0
Turn	20.5	20.5	0
Other	49.7	50.3	0.6

^a^ Results obtained in Bestsel for spectra range 200–250 nm.

**Table 2 ijms-22-02848-t002:** Cas3^W406A^ secondary structure from CD measurements at either side of the inflection point.^a.^

Secondary Structure	% Composition in Cas3^W406A^ at 30 °C	% Composition in Cas3^W406A^ at 35 °C	Change (Δ)
α-helix	22.1	12.8	−9.3
β-antiparallel	22.0	22.8	0.8
β-parallel	6.5	3.7	−2.8
Turn	10.3	11.7	1.4
Other	39.1	49.0	9.9

^a^ Results obtained in Bestsel for spectra range 200–250 nm.

## Data Availability

The data that support the findings of this study are available from the corresponding author upon reasonable request.

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
