# Peer review of "A Tryptophan ‘Gate’ in the CRISPR-Cas3 Nuclease Controls ssDNA Entry into the Nuclease Site, That When Removed Results in Nuclease Hyperactivity"

_ijms, 2021, doi:10.3390/ijms22062848_

Round 1

Reviewer 1 Report

The manuscript by He and collegues represents an appealing attempt to explain T-dependent nuclease activity of E. coli Cas3 protein from a mechanistic perspective.

Cas3 is an ATP-dependent single-strand DNA nuclease-helicase, a key element of the CRISPR-Cas adaptative immunity system that protect against invader genetic elements. CRISPR interference is significantly affected by temperature changes and it has been proposed that Cas3 could expose a switch-like function depending on cell temperature.

Applying a large combination of different techniques, the authors are able to observe important structural differences (CD measurements) upon temperature alterations and ultimately they could correlate the modulation of nuclease function to conformational rearrangements mediated by specific amino acids (W406, W230) predicted to interact with ssDNA at the iHDA1 domain.

The authors have done a great job investigating the reasons of the reported differential activities, offering an interesting mechanistic rationale through the identification of hotspots on Cas3 protein structure responsible for T-dependent activity regulation.

The work is technically robust and provides a remarkable amount of information that could further advance CRISPR systems understanding.

In my opinion there are few concerns that, if addressed, would increase the quality of the manuscript.

1) Overall the figures could be improved, in resolution and labels. I would recommend to label W230 in figure 1A and explicitly show the HD nuclease active site amino acids (may be in C). Figure 3C is scarcely readable. How the authors calculate the hydrophobic interactions mentioned and how these differ between the two panels? From the figure, W230 and W406 belong to alpha helices involved in the structural rearrangement that disrupts the ππ interaction. Could the authors add some comments on that?

2) Although conformational ensemble relies on single MD runs and therefore holds limited statistical significance, the presentation of results could be improved: W406 is reported to stabilize within a pocket in the ssDNA tunnel, it would be useful to add extra reference to this information, for instance they could select surrounding structural motifs and plot key geometric parameters to validate their observation. The authors discuss and graph structures obtained during the final MD simulations: how do they verify their structural representativeness and/or stability?

3) Cas3 3D structure has been obtained using Swiss-model server. They should add sequence and structural alignments details used to obtain the homology model in supporting material.

4) In materials and methods, the description of the MD simulation seems misleading. The authors declare they performed runs at 301K, 303K and 310K for a simulation time between 110 and 300 ns, but this does not correspond to the discussed results, please explain. Importantly, the choice of the Berendsen thermostat, highly discouraged, also needs to be justified.

5) There are many typos, I recommend the authors to carefully check their text.

Author Response

Response to reviewers

We are grateful to the reviewers for their time and for sharing their constructive insights. In the revised manuscript additions are highlighted in yellow and track and change has been utilized within the manuscript to facilitate ease of finding text alteration. Responses to the reviewers comments are shown in below, with key points highlighted in bold. Our response to the comments is item by item in the order we received them:

Reviewer 1:

  1. (a) Regarding the resolution of the figures and labelling of W230 in Figure 1A;

Newer high resolution images have been provided for the figures especially Figure 3C, we are thankful to the Reviewer for noticing it. W230 has been labeled in Figure 2A (as opposed to Figure 1A) as we believe this is the figure the Reviewer was referring to.

               (b) How did the authors calculate the hydrophobic interactions mentioned and how these differ between the two panels;

Hydrophobic interactions during the simulations were monitored by measuring the distances and the radius of gyration of the hydrophobic groups involved in such interactions. Graphs are added into the supporting material as Figure S5. 

               (c) W230 and W406 belong to alpha helices involved in the structural rearrangement that disrupts the ππ interaction;

As the Reviewer correctly noticed, amino acids W406 and W230 belong to different alpha helices. In some simulations the brakeage of π-π interaction between W406 and W230 is indeed followed by the movement of these helices, but that was not the case in all simulations. In some simulations, the movement of these helices is minor and not significant, but the brakeage of π-π interaction between W406 and W230 still occurs. Therefore, we did not dare to draw any conclusions regarding the relationship between the disruption of π-π interaction between W406 and W230 and the movement of helices to which these amino acids belong to. Instead, we focus only on disruption of W406 -W230   π-π interaction regarding the temperature which was confirmed by all performed simulations.

  1. 2. Regarding the final MD simulations and the verification of their structural representativeness and/or stability;

Stability of the systems was monitored through the RMSD (Root Mean Standard Deviation) of the protein backbone during the simulations. RMSD graphs of the simulated systems are included in the supporting material as Figure S7. As mentioned in the answer to the previous remark, only the results that were observed in all simulations are presented in the paper since these results were considered as representative and statistically significant. Key geometric parameters to validate stabilisation of the W406 in ssDNA tunnel are plotted and shown in the supporting material as Figure S5. We measured distances from W406 and hydrophobic parts of a pocket within which W406 is stabilized, as well as radius of gyration formed by W406 and hydrophobic parts of a pocket. In case of protein-DNA simulations, such graphs could not be made since the tunnel is occupied by DNA in the starting structure.

  1. 3. Structural alignments used to obtain the homology model;

Structural alignment of the structures used to generate 3D model of E. coli Cas3 is added in the supporting material as Figure S6.

  1. 4. (a) Description of the MD simulations;

We agree with the Reviewer that the Methods part of molecular modelling was not clear enough regarding the systems that were simulated at various temperatures. In order to clarify it, the following sentence is added at lines 758 - 761: “The following systems were subjected to computational simulations: (i) protein complexes with two Mg2+ ions and ATP, (ii) protein (wild type and mutants) complexes with two Mg2+ ions and ssDNA. Each system was simulated at lower (301 K and/or 303 K) and higher (310 K and/or 317) temperature (Table S1 in supporting material).”

The complete overview of the simulated systems, together with their simulation time and temperature, is shown in the Table S3 in the supporting material.

               (b) Justification of the use of the Berendsen thermostat;

The Reviewer has made a valid point regarding the usage of Berendsen thermostat which is recommended for the equilibration phase rather than for the production phase. However, since all the computational results presented in the paper have experimental validation, we believe that the usage of Berendsen thermostat for the production phase does not present an issue in this case.

  1. Typos in the manuscript;

We thank the reviewer for bring this to our attention, the manuscript has been checked and all typos tracked in the document and additions highlighted.

Reviewer 2 Report

This is a very interesting manuscript, describing the identification and characterization of an amino acid residue in the Cas3 nuclease that is both required for normal ability to protect bacteria against phage infection and, whose mutation to another amino acid yields a version of the enzyme that is hyperactive at higher temperatures. Specifically, the wild type Cas3 is more active as a nuclease at 30 C, showing greatly reduced activity at 37 C. This change is reflected in reduced protection of bacteria against lambda phage infection at the higher temperature. One of several conserved Trp residues (W406), when replaced with Ala, yields a hyperactive nuclease that is resistant to temperature shift. However, as the authors show, this nuclease activity alone does not allow the mutant enzyme to protect cells against lambda infection. The authors provide a variety of protein structural analyses to model the wild type and mutant enzymes, including Circular Dichroism, Molecular Dynamics, and comparisons with Cas3 homologues whose crystal structures have already been analyzed. They go on to propose a cogent model whereby W406 provides a gatekeeping role controlling ssDNA entry; while removal of the “gate” leads to hyperactivity, that alone is not sufficient to protect against phage infection, since structurally, this change removes the mutant enzyme from controls (e.g., interaction with Cascade complex and the HtpG chaperone. No major questions or concerns appear. Only minor stylistic or editorial modifications and a few questions.

Minor points:

Results, Page 4, line 115, Chang to “,,,in more detail why Cas3 behaves…”; line 145, change to “for gaining access to the nuclease active site”

Page 5, line 163: Change to “There was no evidence in Electrophoretic Mobility Shift Assays (EMSAs) for …”

Figure 2. This will, of course, need to be clearer for the final version.

Page 7, line 206: Change to “MD simulations at 37”

Page 10, Figure 4 and legend: Need to indicate the N stands for nicked supercoil and L stands for linear; also, the image in the pdf is not clear enough to visualize the supercoiled material (so maybe that needs to be specifically added to the figure).

Page 11, line 293: Change to “hyperactivity had an impact on…”; line 307: Change to “hyperactive nuclease, its expression…” (its does not have an apostrophe here, since this is possessive).

Discussion, Page 12, line 319: Change to “we found that…”; line 332: Change to “consistent with a switch…”; page 13, line 391: Change to “site-directed mutagenesis kit…”; line 396: Change to “till Absorbance at 600 nm (OD600)”

Page 14, line 397: Change to “reached 0.3 and L-arabinose 0.2% (w/v) was added at this point to induce…”; line 414: Assuming 1000 mM NaCl is as meant, Change to “1M NaCl”; line 445: Change to ‘Running buffer contained 0.13 M”

Page 15, line 453: Change to “before directly mixing with…”

Author Response

Response to reviewers
We are grateful to the reviewers for their time and for sharing their constructive insights. In the revised manuscript additions are highlighted in yellow and track and change has been utilized within the manuscript to facilitate ease of finding text alteration. Responses to the reviewers comments are shown in below, with key points highlighted in bold. Our response to the comments is item by item in the order we received them:
Reviewer 2:
1. Minor points (a) typos identified by the reviewer;
We thank the reviewer for highlighting specific mistakes in the text. These have been altered and changes tracked through track and change.
(b) Figure 2;
A higher resolution version of Figure 2 has been made.
(c) Figure 4 and legend;
The legend for Figure 4 has been updated to define L and N markers shown in the figure, lines 963 – 965.
